# Modeling cell migratory persistence through temporal correlations and angular noise

**Ignacio Montenegro-Rojas**[1], **Martín Andaur-Lobos**[1], **Karol Soler-Orozco**[1], **Diego Castelli-Lacunza**[2], **Cristina Bertocchi**[3,4], **Anastasios Matzavinos**[2]*, **Andrea Ravasio**[1]*

**1** Institute for Biological and Medical Engineering, Schools of Engineering, Medicine and Biological Sciences, Pontificia Universidad Católica de Chile, Santiago, Chile, **2** Institute for Mathematical and Computational Engineering, School of Engineering, Pontificia Universidad Católica de Chile, Santiago, Chile, **3** Faculty of Biological Sciences, Pontificia Universidad Católica de Chile, Santiago, Chile, **4** Graduate School of Engineering Science, University of Osaka, Osaka, Japan

* amatzavinos@uc.cl (AM); and.ravasio@gmail.com (AR)

## Abstract

The persistence of cell migration is a fundamental property of motile behavior, enabling cells to maintain directionality while adapting to fluctuations and external cues. This feature underlies essential processes such as development, immune responses, and cancer invasion. Classical mathematical models have offered key insights into directed migration, yet they often neglect temporal correlations arising from cellular mechanisms that stabilize polarity and protrusion dynamics. Here, we introduce an agent-based model based on stochastic differential equations that integrates fractional Brownian motion to explicitly incorporate translational autocorrelation in cell trajectories. We simulate migration as a function of angular reorientation and the strength of correlated noise. In this framework, temporal correlation stabilizes trajectory features inherited from initial conditions, whereas angular reorientation introduces variability that enables transitions between erratic and directed motion. Our simulations show that, unlike models driven by white noise, positive correlation markedly enhances persistence even under strong angular reorientation. Moreover, the combination of $D_r$ and H gives rise to emergent behaviors, particularly in the presence of taxis, where persistence and responsiveness are jointly tuned. These results identify correlated noise as a proxy for intrinsic cellular memory and provide a versatile computational framework to interpret the diversity and complexity of migratory behaviors.

## Introduction

Cell migration is a fundamental biological process for single and multicellular organisms. In humans, it plays a critical role in various physiological and pathological contexts, such as embryonic development, immune surveillance, wound healing,

**Data availability statement:** The codes used for the simulations and data generation, as well as all original simulation outputs, are available in this repository: https://github.com/itmonte-negro/Persistence-Cell-Migration-Project.git.

**Funding:** This work was funded by ANID FONDECYT Regular 1210872 (AR-CB), 1250073 (CB-AR), 1221220 (AM), 1260593 (AR-CB); ANID FONDEQUIP EMQ210101 (AR-CB); Núcleo Milenio SELFO NCN2024_068 (AR-CB). AR and AM are grateful to PUC/ VRI, PUC IIBM, PUC IMC, and the graduate program at IIBM for seed funding and support. The funders had no role in study design, data collection and analysis, decision to publish, or preparation of the manuscript.

**Competing interests:** The authors have declared that no competing interests exist.

and cancer invasion [1–7]. Cells exhibit characteristic modes of migration to adapt to the physicochemical properties of the microenvironment [8–13]. This is achieved by an intricate interplay of cellular properties, which biologists cumulatively refer to as cellular phenotype, and environmental cues such as composition and density of the extracellular matrix (ECM), tissue rheology and geometry, and soluble or ECM-bound signaling factors. Importantly, gradients and asymmetries of these extrinsic cell factors are known to guide directed motion (i.e., taxis) of otherwise random walking cells [14–19]. The inherent complexity of both intracellular processes and environmental cues gives rise to a high degree of heterogeneity in cell migration with different turning rates and speed. Some cell types display persistent motion, while others wander like random walkers, with a variety of intermediate states exhibiting behaviors that blend directional persistence and randomness. Some cells are more susceptible to external inputs, whereas others primarily follow intrinsic regulatory mechanisms. For instance, Dictyostelium and Polysphondylium amoebae exhibit directed motility even in the absence of external cues, likely due to the stability of cellular processes [20]. This diversity in cellular behavior makes modeling and prediction challenging, as it requires careful consideration of both extrinsic factors and intrinsic cellular properties. For instance, one key determinant of migration directionality and speed is the formation and persistence of lamellipodia, exploratory subcellular protrusions at the leading edge of migratory cells [21]. The stability of lamellipodia, in turn, relies on the cells' ability to form stable adhesions with the substrate, signaling cascades that regulate the branched polymerization of actin, and the formation of a contractile actomyosin cable [21]. Similarly, the localization and temporal turnover of a variety of cellular moduli such as the Microtubule Organizing Center (MTOC), the uropod and other subcellular structures, and gradients of actin polymerization and cellular contractility are critical determinants of the process [9,16,21–26].

Mathematical and computational models offer a quantitative tool to capture core principles of migration, from single-cell motility to collective dynamics. These models serve as quantitative frameworks capable of recapitulating the process by starting from a simplified mathematical representation of the system's key characteristics. In particular, agent-based models (ABMs) use a discrete mathematical characterization of individual cells (agents) and their interactions with the environment to represent and interconnect extrinsic and intrinsic moduli of the process [27–31]. Moreover, incorporation of stochastic properties into individual agents allows to effectively capture the characteristics of migratory cells including the population heterogeneity observed during cancer invasion. Most recent models describe cells as random walkers, where changes in directionality are modeled using a standard Gaussian distribution of possible orientations in the subsequent step. This approach provides a useful baseline for capturing stochastic aspects of migration. Yet, these models do not account for the intrinsic ability of cells to stabilize their movement over different time periods [20,32,33]. The assumption that the random motility of cells follows classic Brownian dynamics does not always align with experimental observations. Single-cell trajectories frequently display anomalous diffusion, reflected in a non-linear time dependence of the mean squared displacement (MSD). Under different

conditions, cell motility can exhibit either subdiffusive or superdiffusive behavior, often reflecting cytoskeletal dynamics, intermittent propulsion, or temporal correlations (¨memory¨) effects in the direction of motion [34,35]. In this context, subdiffusive behavior corresponds to hindered spreading over time, whereas superdiffusive behavior reflects enhanced spreading typically associated with persistent, directed motion. These anomalous diffusive behaviors have been reported across diverse cell types, including fibroblasts [36], epithelial [37] and amoeboid cells [38,39], as well as cancer [40,41] and immune cells [42,43], both in vitro and in vivo. The cellular mechanisms underlying these behaviors operate over finite temporal scales associated with the assembly and disassembly of subcellular structures that sustain directed migration, including lamellipodia, adhesion complexes, and actin networks, in a cell-specific manner [36-39,42,43]. The energetic cost associated with remodeling or maintaining these structures makes it more likely that cells retain a given conformation, thereby stabilizing their migratory machinery [44,45]. As a result, the persistence of these dynamics introduces temporal correlations in cell displacement, commonly referred to as ¨memory¨ in the direction of motion. Current mathematical descriptions of migration tend to overlook these dynamics and, in particular, do not explicitly account for subcellular turnover as a source of temporal correlation underlying persistent motion [46,47]. Nonetheless, these models offer a strong foundation for further refining our understanding of the mechanisms that regulate the persistence of cell migration. In particular, Smeets *et al.* [34] proposed a model based on physical parameters that correspond to measurable biological properties to investigate cell aggregates and their collective dynamics. This proved to be an analytical tool capable of predicting transitions characteristic of, e.g., cancer progression and embryonic development, thus providing a mechanistic understanding of complex processes and of the underlying physical mechanisms. Additionally, this framework is scalable and modular, allowing decomposition into individual components and their systematic analysis. The model is based on stochastic differential equations (SDEs) where the direction of agent movement is regulated by Gaussian noise scaled by the diffusion coefficient ($D_r$); and cellular properties influencing motion and aggregation are mediated by interactions between agents (adhesive and cohesive energies, contact inhibition of locomotion). However, at the single cell level, the model does not account for how intrinsic cell properties, such as cell polarity, can affect migration speed and directionality, disregarding the stabilization of the machinery to achieve a persistent motion [22,35, 40-41]. One stochastic model for capturing such correlated motion is fractional Brownian motion (fBM) [46,47], a non-Markovian, Gaussian stochastic process characterized by correlated increments, originally introduced by Kolmogorov [44] and later formalized by Mandelbrot and van Ness [45,48]; that has been widely applied across disciplines, including economics and social systems [49,50]. In contrast to models based on uncorrelated noise (Langevin equations; continuous random walk, etc.), fBm allows the introduction of long-range temporal correlations while maintaining stationary increments [46,51]. The increments in an fBm function are not independent but influenced by prior states (hence, the non-Markovian characteristic). This correlation can be modulated to introduce positive or negative feedback in the dynamics and can be efficiently simulated using numerical methods such as fractional integration of white noise [52–54], exhibiting a wide array of non-trivial, emergent behaviors [55,56]. Recent studies have successfully fit fBM models to different biological processes [57,58], and to cell migration data, using metrics such as MSD scaling, velocity autocorrelations, and trajectory segmentation to infer time-dependent persistence [59,60]. In particular, the anomalous motility of epithelial cells [61], neutrophils [62], and Drosophila hemocytes [60] has been quantitatively described using fBM or related fractional models. These observations suggest that temporal correlations in spontaneous cell migration, even in the absence of external cues, may contribute to the emergence of tissue-scale patterns [63,64].

## Results

### Stochastic correlation through fractional Brownian motion

Formally, fractional Brownian motion with Hurst index $H \in (0, 1)$ is a centered Gaussian process $W^H(t)$ with continuous paths and stationary increments, characterized by the relation $E\left[\left(W^H(t) - W^H(s)\right)^2\right] = |t - s|^{2H}$. For all $t, s \geq 0$. It follows

immediately that when $H = 1/2$, fractional Brownian motion coincides with standard Brownian motion [65–67]. Moreover, the process exhibits superdiffusive behavior when $H > 1/2$ and subdiffusive behavior when $H < 1/2$. A straightforward algebraic manipulation of the above equation yields that for all $H \in (0, 1)$:

$$Cov\left(W^H(t), W^H(s)\right) = E\left[W^H(t)W^H(s)\right] = \frac{1}{2}\left(t^{2H} + s^{2H} - |t - s|^{2H}\right)$$

Assuming $t > s > 0$ and that $W^H(0) = 0$, we then obtain

$$E\left[\left(W^H(s) - W^H(0)\right)\left(W^H(t) - W^H(s)\right)\right] = E\left[W^H(t)W^H(s)\right] - E\left[\left(W^H(s)\right)^2\right] = \frac{1}{2}\left(t^{2H} - s^{2H} - |t - s|^{2H}\right)$$

Note that the right-hand side of this expression is positive when $H > 1/2$ and negative when $H < 1/2$. Therefore, the increments of fractional Brownian motion are positively correlated when $H > 1/2$ and negatively correlated when $H < 1/2$. For this work we restrict our analysis to migratory dynamics with $H \geq 1/2$, modeling the temporal stability of biological structure associated with persistent motion. Negative correlation ($H < 1/2$) lacks a clear biological interpretation in this context, and it is therefore not considered.

### Derivation of a correlated model for cell migration

In our model, cells are described as self-propelled particles whose motion is derived from Smeets *et al.* [34].

$$F_m \hat{p} = \gamma_s \frac{dx}{dt}$$

and where variation in the angle of motion is described as

$$\frac{d\theta}{dt} = \sqrt{2D_r}\xi$$

This derivation, which, unlike the original, does not consider cell-cell interactions and adhesive energies, gives rise to motility with constant increments and directionality governed by a diffusion coefficient ($D_r$) and associated Gaussian noise ($\xi$). This model does not account for the energetic costs associated with altering biological processes, such as lamellipodia dynamics, that underline the characteristic temporal correlations and variability in both the magnitude and directionality of cell migration. To the best of our knowledge, no existing model captures these energetically constrained mechanisms. To address this, we propose incorporating translational noise in the equation of motion and generalizing the stochastic term by describing it as correlated noise using fBm theory. Instead of assuming simple Gaussian distribution, variability is defined as $\frac{dW^H}{dt}$, where H describes the magnitude of correlation. Thus, the equation of motion for a single cell that includes stochastic noise is

$$F_m \hat{p} = \gamma_s \frac{dx}{dt} - \alpha \frac{dW^H}{dt}$$

(1)

where the left-hand term is the intrinsic cell motile force of magnitude $F_m$ and direction vector $\hat{p}$, and, on the right side, $\gamma_s$ is the substrate friction coefficient and $\frac{dW^H}{dt}$ is the term describing the correlated noise of magnitude $\alpha$. The term $\alpha\frac{dW^H}{dt}$ represents the translational noise that defines the variation in the increments and it is independently defined for its x and y components ($\alpha\frac{dW^H y}{dt}; \alpha\frac{dW^H y}{dt}$). The parameter $\alpha$ was determined to be 0.25, as this value balances the magnitude of the

translational noise with the constant magnitude of the direction vector multiplied by the timestep (S1 File). On the other hand, the function for the change of orientation is

$$\frac{d\theta}{dt} = \sqrt{2D_r}\frac{dW^{(H_\theta)}}{dt}$$

(2)

where the degree of correlation in $\frac{dW^{(H_\theta)}}{dt}$ is $H_\theta = 0.5$ to model Gaussian white noise. The solution for $\theta[t] = \theta[t - \Delta t] + \Delta\theta$ is then fed into the equation of motion as the direction vector $\hat{p} = (cos\theta, sin\theta)^T$ to solve for the increment $\Delta x$. Thereafter, the position is updated to $x[t] = x[t - \Delta t] + \Delta x$. This solution considers two stochastic effects, white noise in the orientation angle, and correlated noise in the position that affects the increment (S1 File).

## Numerical simulations

Numerical solutions for varying parameter values allow us to evaluate the effect of different levels of positively correlated noise ($H$) on the agent motility in combination with variations in the diffusion coefficient ($D_r$). For each set of $H$ and $D_r$ parameters, we conducted 200 simulations of isolated cells. The initial orientation $\theta$ was randomly assigned within $[0, 2\pi]$, ensuring circularly distributed trajectories. Angular and translational noises were initialized using values sampled from a Gaussian distribution $\sim \mathcal{N}(0, 0.1)$, enabling the assessment of $H$ and $D_r$ on a population of agents with heterogeneous starting conditions. Each simulation comprised 1000 timesteps of size $\Delta t = 0.1$, totaling 100 time units. Cell motile force $F_m$ and friction $\gamma_s$ were given an arbitrary value of 1 across all simulations. To relate the simulated trajectories to biologically relevant behaviors, we analyzed the persistence factor ($PF$), relative turning angle ($\theta_R$), and increment ($|\Delta x|$) for each simulation. The PF was defined as the ratio between the displacement and the total distance traveled by each individual cell:

$$PF = \frac{\left\|\overrightarrow{x_f} - \overrightarrow{x_0}\right\|}{\sum_{i=t_0}^{t_f-\Delta t} \left\|\overrightarrow{x}_{i+\Delta t} - \overrightarrow{x}_i\right\|}$$

(3)

where $\overrightarrow{x_f}$ and $\overrightarrow{x_0}$ are the positions of the agent at the beginning and the end of the simulation defining the total displacement, and $\sum_{i=t_0}^{t_f-\Delta t} \left\|\overrightarrow{x}_{i+\Delta t} - \overrightarrow{x}_i\right\|$ is the total distance travelled (sum of steps). With this definition, the persistence factor is constrained between 0 and 1, where 1 represents maximum persistence (a perfectly straight trajectory). The relative turning angle, which quantifies the changes in the migratory pattern, was defined as the angle between three consecutive cell positions $x_1, x_2, x_3$, where we name the distances between the points as $a$ for distance between $x_1$ and $x_2$, $b$ for distance between $x_2$ and $x_3$, and $c$ for the distance between $x_1$ and $x_3$. Thus, the relative turning angle $\theta_R$ is defined as:

$$\theta_R = \pi - arccos\left\{\frac{(a^2 + b^2 - c^2)}{2ab}\right\}$$

(4)

where $\theta_R$ ranges between $[0, \pi]$. The relative turning angle is 0 radians (no change in directionality) when $x_1, x_2,$ and $x_3$ are aligned on a straight line, and $\pi$ when the agent inverts its direction. Finally, we quantified increment as the magnitude of the vector between two consecutive points. From Equation [1] it that the increment depends on both stochastic effects, as in the following expression:

$$|\Delta x| = \sqrt{\Delta t^2 + 2\alpha\Delta t\left(cos\theta\Delta W_x^H + sin\theta\Delta W_y^H\right) + \alpha^2\left[\left(\Delta W_x^H\right)^2 + \left(\Delta W_y^H\right)^2\right]}$$

(5)

## Analysis of isolated noises: deterministic and single-noise simulations

We first examined the case where neither mechanism was active ($D_r = 0$ and $\alpha = 0$). This deterministic model serves as a benchmark for comparing the relative contribution of angular diffusion and translational noise. Visual representations showed that all simulations were characterized by the same migratory pattern, with all cells deterministically migrating on straight trajectories and covering the same distance (Fig 1A; $D_r = 0$, $\alpha = 0$). Consequently, the plot of the simulations' final position unveiled a circular distribution at a fixed distance from the origin of 100 length units in radius, which serves as reference for all conditions, as result of the 1000 timesteps with $\Delta t = \Delta x = 0.1$. The absence of translational noise and angular diffusion determined each simulation to progress in straight lines with constant increments and no variation in relative angles (Figs 2 and 3; $D_r = 0$, $\alpha = 0$), resulting in constant trajectory features across simulations and timesteps (Fig 1B-D, Fig 2B, C and Fig 3; $D_r = 0$, $\alpha = 0$).

Next, we evaluated the role of angular diffusion ($D_r$) in the absence of translational noise. This was achieved by setting $\alpha = 0$ in the equation of motion. Variation of $D_r$ alone produced migratory patterns characteristic of stereotypical random walkers (Fig 1A; $D_r = 0.1 - 10$, $\alpha = 0$) with increments being constant at any given temperature of the system ($\Delta x = \Delta t = 0.1$ – Fig 2A, B and Fig 3A; $D_r = 0.1 - 10$, $\alpha = 0$). Consequently, the total distance traveled by the agents (sum of all increments) remained constant across all $D_r$ values (Fig 1B; $D_r = 0.1 - 10$, $\alpha = 0$). However, as $D_r$ increased and directional changes became more frequent (Fig 2A, C and Fig 3B; $D_r = 0.1 - 10$, $\alpha = 0$), the endpoints of the trajectories

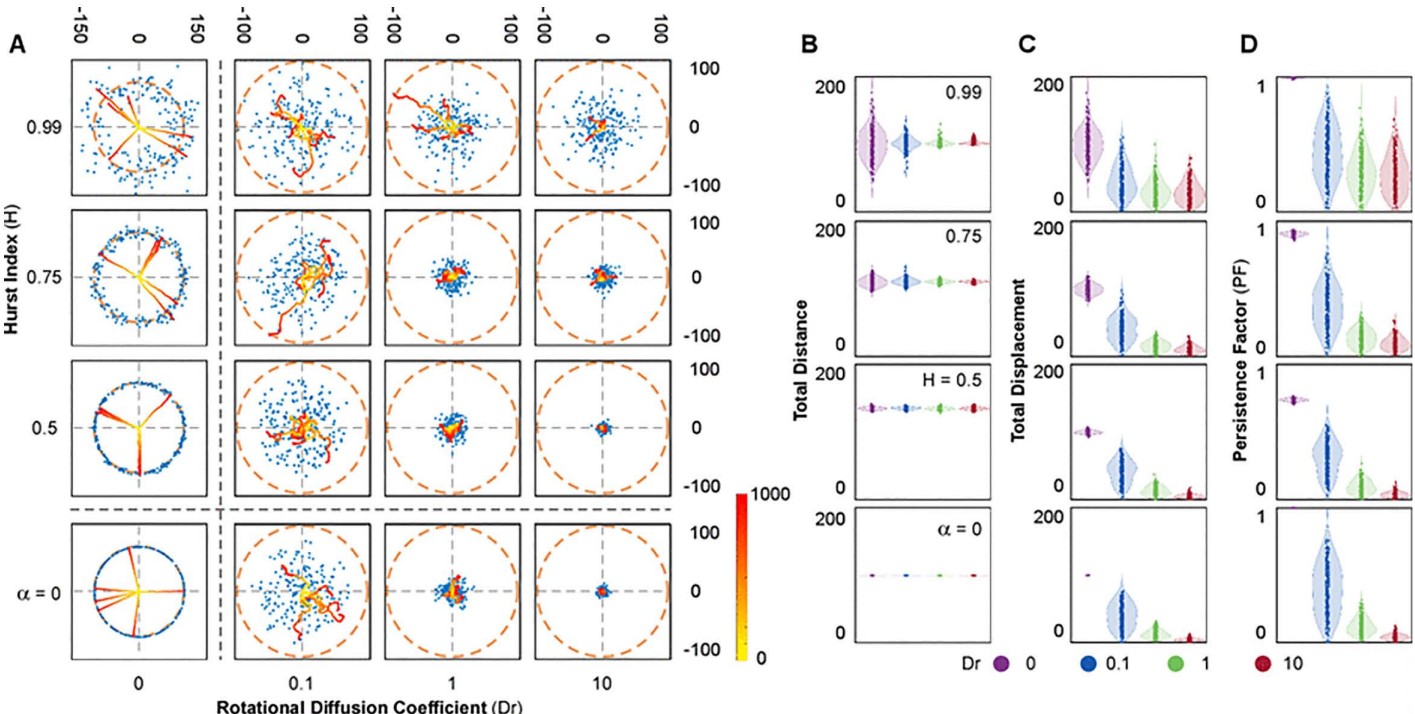

**Fig 1. Long-scale behaviour of simulated cells. (A)** Representative simulations and final position distributions for various combinations of persistence parameter (H) and rotational diffusion (Dr). Trajectories from six randomly selected simulations are displayed, with their 1000-step time evolution indicated by a yellow-to-red colormap (see also Suppl. Movie 1 for the trajectory dynamics). Final positions from 200 simulations are marked as blue dots. The dashed orange circle denotes the final position for the deterministic model (α = 0, Dr = 0) as a reference. **(B–D)** Distributions of total distance traveled **(B)**, total displacement **(C)**, and persistence factor (D) across different values of H and Dr. From top to bottom, each plot corresponds to a decreasing H value, while colours represent different Dr levels. Dot plots overlaid on violin plots depict the distribution of outcomes across simulations.

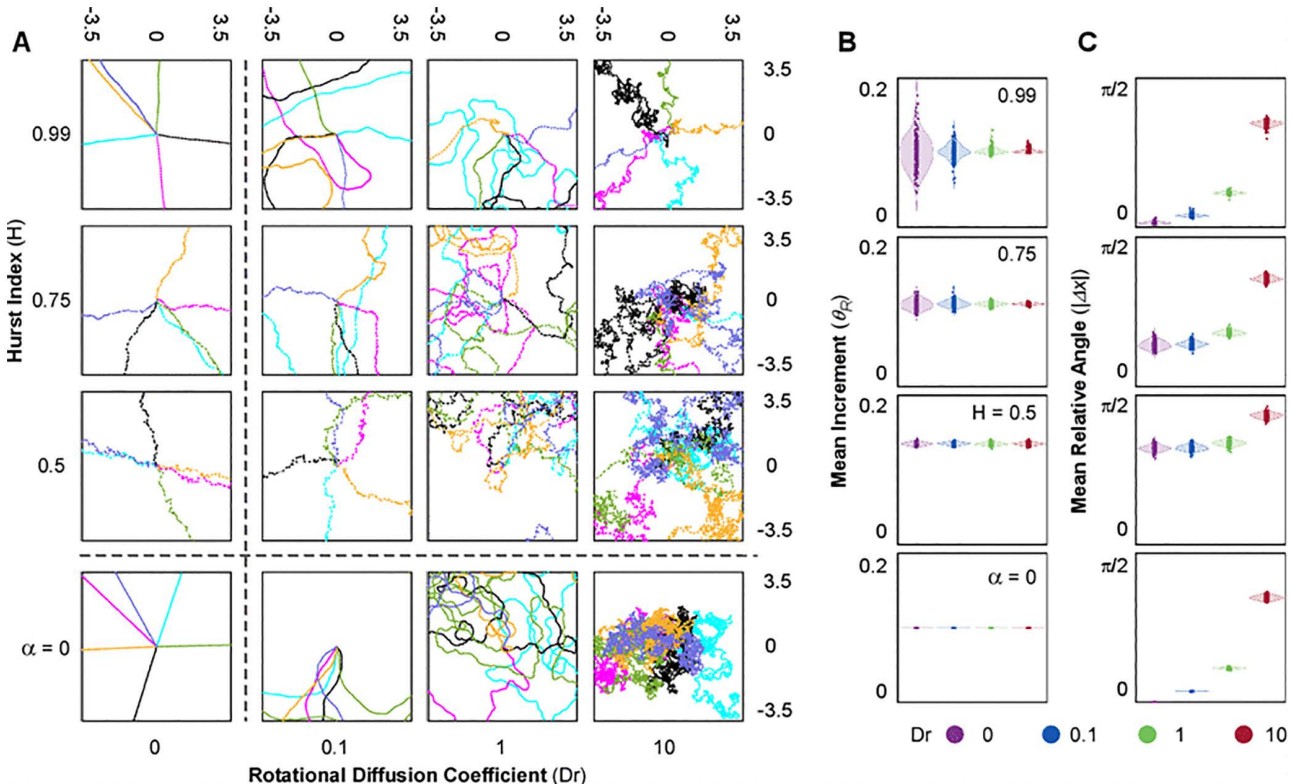

**Fig 2. Short scale behaviour of simulated cells. (A)** Short scale representation of trajectories. Zoomed Trajectories. Each plot consists of a focus of 6 simulated trajectories in the space surrounding the origin. Each position x[t] is represented as a point, with each color representing one individual simulation. **(B and C)** Distribution of Increment and Relative Angle (quantified in x axes). Each Dr value is represented in a different color. The deterministic model ($\alpha$=0, bottom row) has a constant increment $\Delta$x = 0.1, and quantifications for $\theta_R$ are in radians ranging from [0, 2$\pi$].

distributed closer to the origin (Fig 1A, B; $D_r$ = 0.1 – 10, $\alpha$ = 0). Reduced displacement despite a constant total distance traveled led to a decrease in persistence (Fig 1D; $D_r$ = 0.1 – 10, $\alpha$ = 0).

On the other hand, we modeled the effect of translational noise by testing three values of the Hurst Index in the equation of motion in the absence of angular diffusion ($D_r$ = 0): $H$ = 0.5, representing Gaussian white noise, and two positively correlated cases, $H$ = 0.75 and $H$ = 0.99. These simulations resulted in trajectories displaying high directionality and final positions distributed along the reference circle characteristic of the deterministic model (Fig 1A; $D_r$ = 0, $H$ = 0.5 – 0.99). Interestingly, the endpoints for simulations with Gaussian noise ($H$ = 0.5) were distributed close to the reference circle, whereas increasing the degree of correlation ($H$ = 0.75 and 0.99) amplified the variability in this distribution. Since the translational noise is defined by two independent directional components ($\alpha\frac{dW^H x}{dt}$; $\alpha\frac{dW^H y}{dt}$), the resulting vector introduced short-scale fluctuations in both increment and directionality. When translational noise is modeled as white noise ($H$ = 0.5), both increments and relative angles were randomly perturbed at every timesteps, resulting in ¨noisy¨ displacement patterns as compared to the deterministic model (Fig 2A-C and Fig 3; $D_r$ = 0, $H$ = 0.5). Analysis of total distance and total displacement further reflected this minor deviation. While the increased fluctuations slightly elevated the total distance traveled, the total displacement remained unchanged, resulting in a marked decrease in the persistence factor (Fig 1B-D; $D_r$ = 0, $H$ = 0.5). In contrast, when positive correlation was introduced ($H$ = 0.75 and 0.99), trajectories became more stable, as the fBm reduced the variation in increment and relative angle over time within each simulation (Fig 2A and Fig 3; $D_r$ = 0, $H$ = 0.75 – 0.99). However, the analysis of the distribution of these parameters across different simulations showed

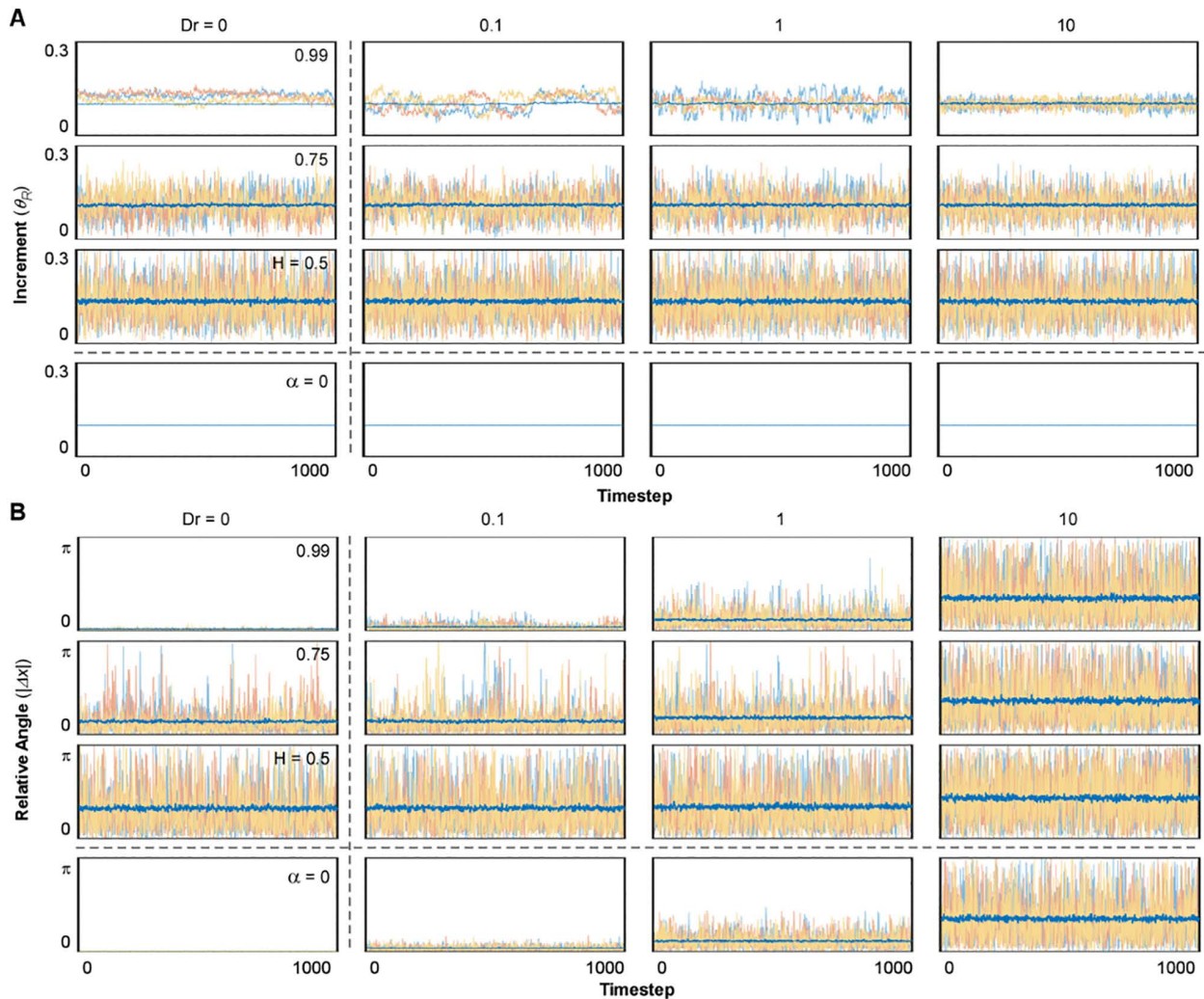

**Fig 3. Time evolution of short scale quantities. (A) Time evolution of increment and (B) Time evolution of relative angle.** Each plot represents 3 exemplary simulations (coloured yellow, red, and blue) for each timestep of the simulation. The average value of the 200 simulations for each timestep is represented in a thicker, dark blue line overlapped with the individual simulations (with transparency). The range of values is higher than that of previous representations, going from [0, 0.3] for $\Delta$x and from [0, $\pi$] for $\theta_R$.

increased variability (Fig 2B, C; $D_r$ = 0, H = 0.75 – 0.99), as a consequence of amplified variability in the randomized initial conditions (Fig 4; $D_r$ = 0, H = 0.75 – 0.99). This led to greater variability in the distributions of total distance and total displacement (Fig 1B-C; $D_r$ = 0, H = 0.75 – 0.99). Notably, while the average total distance decreased, the average total displacement increased. Together, these effects resulted in a pronounced rise in trajectory persistence, with the persistence factor approaching 1 for H = 0.99 (Fig 1 D; $D_r$ = 0, H = 0.75 – 0.99).

### Synergistic effects and emergent dynamics from combined stochastic mechanisms

To analyze the nonlinear interaction between angular diffusion and correlated translational noise, we tested three values of $D_r$ (0.1, 1 and 10) and three values of H (0.5, 0.75 and 0.99). When angular diffusion was varied in the presence of uncorrelated translational noise (H = 0.5), the resulting trajectories closely resembled, at the large-scale, those observed

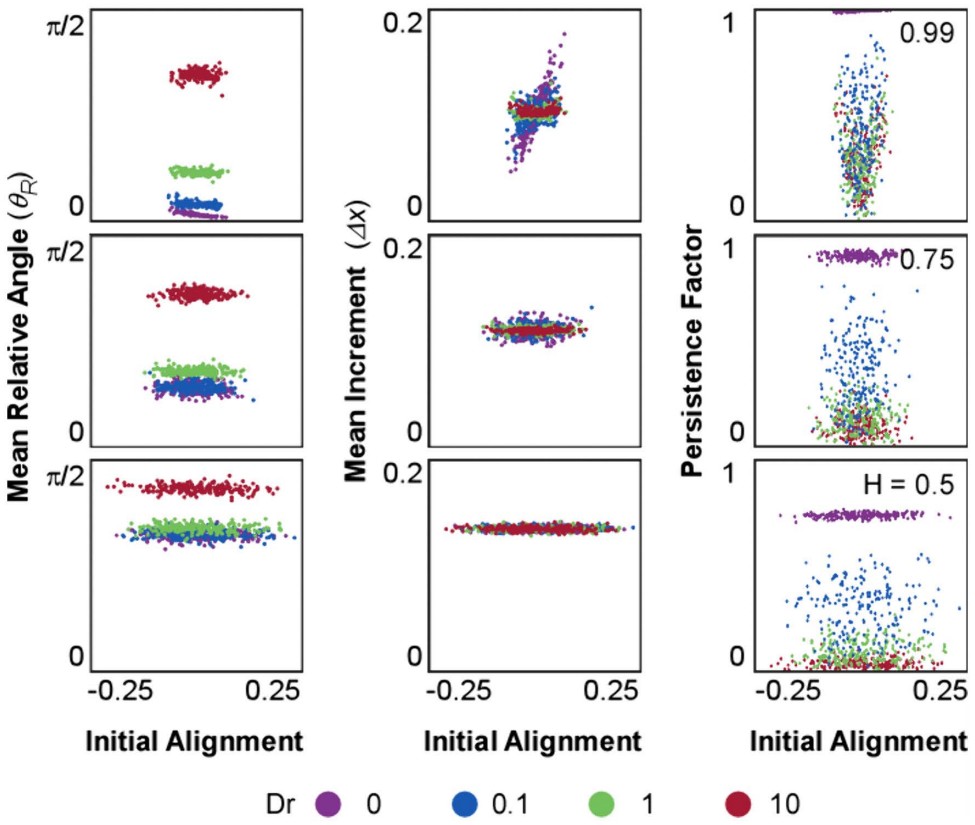

**Fig 4. Analysis of initial orientation.** Each plot shows a scattering comparing the initial alignment between the correlated noise and the orientation vector p̂ with the relative angle($\theta_R$, left); increment ($\Delta x$, middle), and Persistence Factor (PF, right). Each plot represents the quantities coloured by the value of Dr (labeled at the bottom). From top to bottom the value of H decreases (labelled at the right), and because the condition $\alpha = 0$ has no noise component, it is not shown. Quantifications for $\theta_R$ are in radians and the Initial Alignment is defined in the range between [-$\alpha$, $\alpha$].

in the absence of translational noise ($\alpha = 0$) and were mostly dominated by $D_r$ (Fig 1A; $D_r = 0.1 - 10$, $H = 0.5$). Interestingly, compared to $\alpha = 0$, Gaussian translational noise introduced variability and increased the magnitude of displacement increments (Fig 2A, B and 3A; $D_r = 0.1 - 10$, $H = 0.5$). This suggests that even in the absence of angular instability, uncorrelated translational noise alone can modulate increment and trajectory curvature. Relative angles were also larger and more variable. However, $D_r$ modulation was largely lost, with only $D_r = 10$ producing a substantial increase relative to lower $D_r$ values (Fig 2C and 3B; $D_r = 0.1 - 10$, $H = 0.5$). As result, the total distance traveled increased, while total displacement remained similar to the $\alpha = 0$ condition (Fig 1B, C; $D_r = 0.1 - 10$, $H = 0.5$). This led to a general reduction in PF, particularly at low $D_r$ values (Fig 1D; $D_r = 0.1 - 10$, $H = 0.5$). While the variability of the increment represents an interesting additional feature to the original Smeets formulation, more closely resembling cellular migration, it remains a largely expected outcome of adding a noise term to the equation of motion [1]. However, the incorporation of positive correlation gave rise to emergent behaviors that could not be anticipated *a priori*, as it provides persistence through the stabilization of cellular programs that define migratory patterns, a behavior often referred to as memory. As result, the trajectories appeared to be more heterogenous relative to each other, especially for higher $H$ and lower $D_r$ (Fig 1A and 2A; $D_r = 0.1 - 10$, $H = 0.75 - 0.99$), resulting in increased dispersion of end points relative to $H = 0.5$. This was further reflected in the distributions of total distance traveled and net displacement, which, despite exhibiting only marginal shifts in their mean values, displayed markedly increased variance across simulations, particularly at higher values of $H$

(Fig 1B, C; $D_r$ = 0.1 – 10, $H$ = 0.75 – 0.99), indicating that translational correlation amplifies the influence of initial conditions (Fig 4). Such variability mirrors the heterogeneous motility phenotypes often seen in migrating cells, where internal states and polarity cues differ despite uniform environmental conditions [10,12,32]. Consistent with these observations, both the mean and variance of the persistence factor increased as correlation strengthened (Fig 1D; $D_r$ = 0.1 – 10, $H$ = 0.75 – 0.99). This trend was paralleled by changes in the distribution of increment magnitudes, with greater dispersion across the population of simulations (Fig 2B; $D_r$ = 0.1 – 10, $H$ = 0.75 – 0.99). Notably, while the mean increment across all simulations decreased only slightly as $H$ increased, its variance rose substantially, indicating greater dispersion across the population. This behavior remained largely independent of $D_r$, indicating that translational correlation primarily governs increment dispersion. In contrast, both the distribution and variance of relative turning angles declined substantially at higher $H$ values, but primarily at low to intermediate $D_r$, where translational correlation played a stronger modulatory role. At $D_r$ = 10, however, both the mean and variability of turning angles were largely insensitive to changes in $H$, as the very high angular diffusion was the predominant factor shaping the system's behavior. This insensitivity illustrates how excessive angular reorientation can override intrinsic persistence, aligning with behaviors observed in cells undergoing chemokinetic activation or loss of polarity [23,68,69]. In addition to these global trends, the temporal evolution of individual trajectories further confirmed that both the increment and the relative turning angle were attenuated by increasing $H$ and amplified by higher $D_r$ values over time (Fig 3A, B). This highlights how translational correlation acts as a stabilizing force at short timescales, whereas angular diffusion progressively introduces variability, reinforcing the opposing contributions of these two stochastic mechanisms across dynamic regimes.

Finally, to explore how initial stochastic conditions influence migratory outcomes under different parameter regimes, we examined the relationship between initial alignment and three key metrics: mean relative angle, mean increment, and persistence factor (Fig 4, S1 File). The relative angle remained largely unaffected by translational correlation across all values of $H$, suggesting that directional fluctuations are not directly modulated by temporal correlations. In contrast, at high correlation ($H$ = 0.99), both the mean increment and persistence factor became strongly dependent on initial alignment, indicating that positive correlation amplifies the influence of initial conditions and in turn shapes long-term persistence. This dependency was most pronounced at low $D_r$, where individual simulations exhibited a broad spread in both increment and PF values. As $D_r$ increased, this sensitivity to initial alignment was progressively attenuated or distributed. These findings highlight a nonlinear interaction between $H$ and $D_r$, wherein translational correlation establishes a persistent migration that is either preserved or disrupted by the strength of angular diffusion. This behavior aligns with reported *in vitro* cell migration patterns, where correlation of motion was observed and quantified, revealing distinct migratory forms under different correlation regimes [32,59]. Our results highlight how translational correlation acts as a stabilizing force at short timescales, whereas angular diffusion progressively introduces variability, reinforcing the opposing contributions of these two stochastic mechanisms across dynamic regimes. Variance in persistence factor, total displacement, and increment magnitude increases with $H$, particularly when $D_r$ is low, confirming that translational correlation amplifies the impact of initial stochastic conditions. Meanwhile, the observed reduction in angle variability at higher $H$ and the overriding effect of $D_r$ at high diffusion levels are consistently reflected across simulations. Together, these features demonstrate that the mixed model recapitulates a broad spectrum of migratory behaviors observed in biological contexts, including stable persistence, variable exploration, and trajectory dispersion arising from intrinsic cellular states.

## Mean square displacement analysis

To characterize the anomalous diffusive behaviors observed in our simulations we calculated the Mean Square Displacement (MSD, Fig 5). This analysis allows us to characterize the anomalous diffusion the fBm introduces into the migratory cells. We define the MSD as the ensemble average of the squared total distance:

$$MSD = \left\langle |x(t) - x_0|^2 \right\rangle = \frac{1}{N} \sum_N |x^i(t) - x_0|^2$$

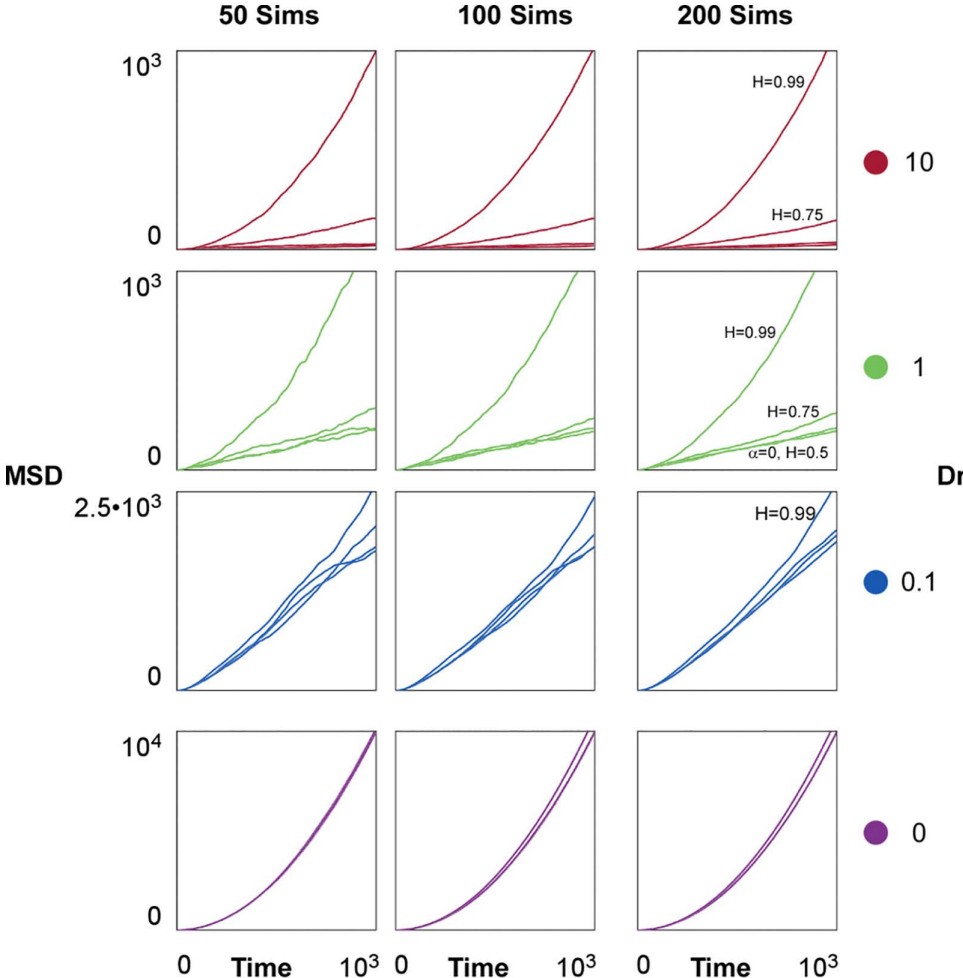

**Fig 5. Mean Square Displacement (MSD) Analysis.** Each plot shows 4 distinct curves, representing the MSD for each different level of correlation ($\alpha = 0$; $H = 0.5, 0.75, 0.99$). Color code is based on the value of $D_r$, increasing from bottom to top. Each MSD curve considers the average of different simulation sample-sizes for each condition (from left to right; 50, 100, and 200 simulations used), plotted over time. For the last column that considers all simulations, the different curves are labelled when appropriate. Scale is adjusted depending on the $D_r$ value (left axis).

The MSD is defined solely by the final position at the measured time $x(t)$ for each cell, as they are seeded at the origin ($x_0 = 0$). We analyze not only the type of diffusion in the system, but also the stability of the MSD. To this extent, we varied the sample-size $N$ between 50, 100, and 200 samples (25, 50, and 100% of the total simulations made). This stability analysis showed that the high variance introduced into the system by our two stochastic components reflects in instabilities in the MSD curves, yet by sampling over 50% of all simulations we can adequately recount the global behavior (Fig 5; sample size 100 and 200 simulations). We analyze the type of diffusion based on the full set of simulations, using all 200 simulations for each parameter combination.

To characterize the anomalous diffusion, we analyze qualitatively the evolution of the MSD over time. For Brownian Motion (uncorrelated noise), the MSD evolves as a linear relation with time ($MSD \propto t$), whereas it scales proportional to a power law $MSD \propto t^\alpha$ for other cases, with $\alpha < 1$ for subdiffusive and $\alpha > 1$ for superdiffusive processes. When no reorientation is present (Fig 5; $D_r = 0$) we note that the process isn't pure diffusion (as seen in typical MSD curves), as there is still a constant advection term that contributes due to $F_m$, which produces a constant velocity causing ballistic MSD

curves. Interestingly, as angular diffusion increases, there are noticeable differences in the MSD, which seem to depend on the magnitude of correlation (Fig 5; $D_r$ = 0.1, 1, 10). Highly correlated traces ($H$ = 0.99) exhibit superdiffusion irrespective of the value of $D_r$, only varying in the magnitude of the MSD (almost exactly a tenfold between $D_r$ = 0 and $D_r$ = 10). Variance in the magnitude is related to the displacement and total distance analysis (Fig 1B-C, $H$ = 0.99), where we observe higher displacements for positive correlation.

These differences are more evident at higher values of $D_r$ (Fig 5; $D_r$ = 1, 10), where correlation seems to modulate the behavior from diffusive to superdiffusive, with reorientation playing a role in regulating the total displacement achieved (magnitude of the MSD). When in absence of the correlation module and with uncorrelated Brownian noise (Fig 5; $Dr$ = 1, 10; $\alpha$ = 0; $H$ = 0.5), the MSD ressembles diffusive behavior, with linear tendencies of the MSD which appear distorted due to the advection component. Our findings align with *in vitro* traces, where the MSD for cells migrating on two-dimensional substrates also exhibits superdiffusive tendencies. It remains of interest to perform a fitting of power curves, to characterize this exponential evolution of the MSD.

## Interplay between combined stochastic mechanisms and directional guidance

Next, we examined how the interaction between angular diffusion and correlated noise affected directional guidance (i.e., taxis). While various taxis mechanisms and models have been described [15], including those based on gradient sensing or polarity alignment, we opted for a minimal implementation suitable for testing the interaction with stochastic motility. In this model, an organizing center exerts a directional influence defined by the angle between cellular orientation and the vector toward the target. This approach integrates naturally into our framework while remaining adaptable to more complex guidance schemes. More complex taxis models can also be integrated in future work or tailored to specific biological scenarios. In the present formulation, directional guidance enters exclusively through the angular equation, which becomes:

$$\frac{d\theta}{dt} = \sqrt{2D_r}\frac{dW^{(H_\theta)}}{dt} - f_{tax} \cdot arccos\frac{v^{org} \cdot \hat{p}}{||v^{org}||}$$

(6)

Where $f_{tax}$ is the strength of the taxis interaction and the term $arccos\frac{v^{org} \cdot \hat{p}}{||v^{org}||}$ represents the difference between the desired direction towards $x_{org}$ and the direction the cell is migrating at any given timestep (S1 File). We performed 100 simulations placing the organizing center at a fixed distance of 100 units from the origin ($x_{org} = \left(-100/\sqrt{2}, -100/\sqrt{2}\right)^T$), on the reference ring defined by the deterministic system (Fig 6, $D_r$ = 0, $\alpha$ = 0). We varied both $H$ and $D_r$ as in the previous analysis and tested three levels of attraction strength ($f_{tax}$): 0.01, 0.1 and 1.

We first examined the behavior of cells in the deterministic case ($D_r$ = 0, $\alpha$ = 0) in the presence of a directional cue. In this regime, the absence of stochasticity results in trajectories that maintain constant orientation and magnitude throughout the simulation (constant distance traveled – Fig 6A; $D_r$ = 0, $\alpha$ = 0).Thus, the motion of cell strongly depends on the initial orientation relative to the organizing center and the strength of the taxis cue (Fig 6; $D_r$ = 0, $\alpha$ = 0). When this initial orientation is aligned with the organizing center, cells proceed directly toward the target. Others gradually swirl and curve as they progress, resulting in a distribution that becomes increasingly biased toward the center with increasing $f_{tax}$. Cells initially oriented opposite to the organizing center traveled in straight paths and accumulated on the reference ring in positions opposite the target. Although some cells escape the attraction, particularly at low $f_{tax}$, the majority move toward the target, as confirmed by the reduced final distance to the organizing center (Fig 7B; $D_r$ = 0, $\alpha$ = 0). Similarly to the deterministic condition, the effect of the external signal under angular diffusion ($D_r$ = 0.1 – 10, $\alpha$ = 0) was strongly influenced by both the initial orientation and the strength of the taxis cue. As in the deterministic case, the total distance traveled remained constant (Fig 7A; $D_r$ = 0.1 – 10, $\alpha$ = 0). However, the introduction of angular diffusion allowed cells to explore alternative directions and gradually reorient toward the organizing center especially at high $f_{tax}$ (Fig 6; $D_r$ = 0.1 – 10, $\alpha$ = 0).

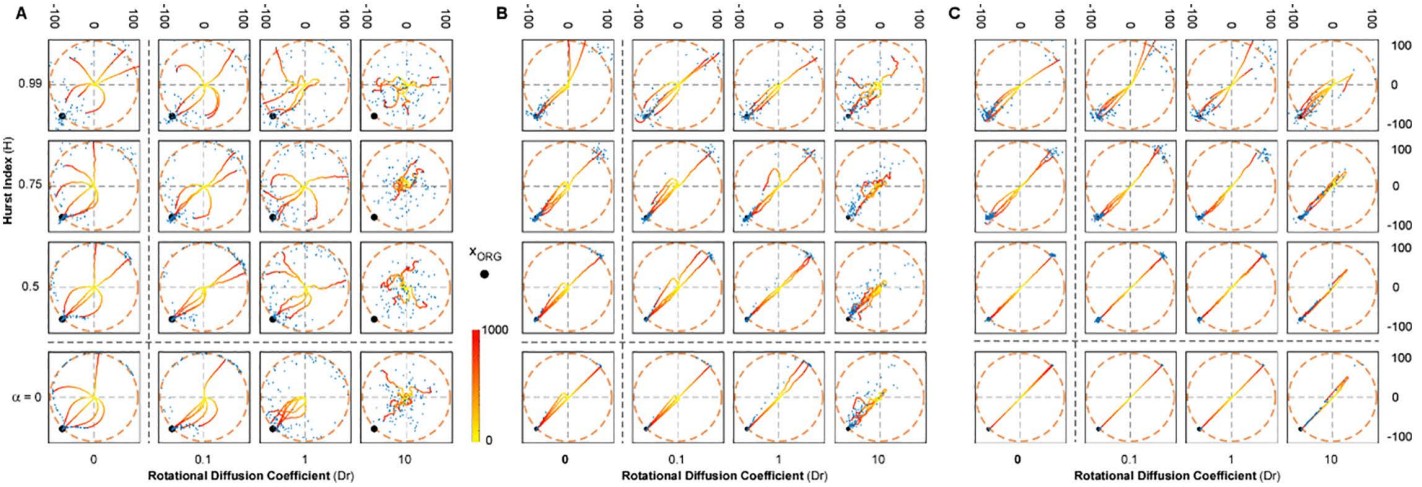

**Fig 6. Trajectories of environmentally guided cells.** Each grid position shows 6 simulated trajectories for different combinations of H and Dr overlapped with a Final Position Scattering, showing the final position of each cell as a blue dot. In each plot an orange ring is added representing the deterministic control model ($\alpha$=0 and Dr=0); and a black dot represents the position of the organizing center $x_{org}$. From left to right each grid shows a different magnitude of attractive force (low $f_{tax}$ = 0.01 on the left, $f_{tax}$ = 0.1 in the middle, and high intensity $f_{tax}$ = 1 on the right). Suppl. Movies 2–4 display the temporal evolution of these trajectories under the three taxis strengths, illustrating how the interplay of correlated noise (H) and angular diffusion (Dr) enables cells to align, overshoot, or escape the organizing center.

This enhanced the chances of alignment with the cue, particularly at lower $D_r$ values, whereas, when $D_r$ is too high ($D_r$ = 10), orientation changes become overly frequent and undermines the taxis mechanism. Low $D_r$ increased the number of cells capable of reaching the organizing center by enabling gradual reorientation without fully disrupting directional migration (Fig 7B; $D_r$ = 0.1, $\alpha$ = 0). However, at higher $D_r$ ($D_r$ = 1 – 10), frequent orientation changes became counter-productive: cells were unable to sustain a directed path, leading to broader spatial distributions and increased variance in distance to the target. We then investigated the isolated effect of translational noise without angular diffusion ($D_r$ = 0, $H$ = 0.5 – 0.99). At $H$ = 0.5, cells displayed minor deviations from straight trajectories even when initially oriented opposite to the organizing center. Those not initially oriented opposite to the organizing center smoothly reoriented toward the target over time (Fig 6; $D_r$ = 0, $H$ = 0.5). Increasing $H$ led to broader spatial distributions, especially at low $f_{tax}$, as positive correlation enhanced cell persistence and increment magnitude (Fig 6; $D_r$ = 0, $H$ = 0.75 – 0.99). This enabled some cells to overshoot the target. Analysis of the distributions confirmed that high $H$ and $D_r$ introduced considerable variance in final positions and distance to the target, even at high taxis strength. Finally, we introduced white noise ($H$ = 0.5) and positively correlated noises ($H$ = 0.75 – 0.99) combined with angular diffusion ($D_r$ = 0.1 – 10). Under Gaussian translational noise and low $f_{tax}$, most cells reoriented toward the organizing center, though with greater variability compared to angular noise alone. Fig 6 and 6B demonstrated increased convergence towards the target for all $D_r$, with a more pronounced effect at high $f_{tax}$. Correlated noise ($H$ = 0.75 – 0.99) resulted in migration dynamics shaped by the interplay between intrinsic persistence and external guidance. At low $f_{tax}$, high $H$ reduced sensitivity to the directional cue, allowing some cells to maintain their initial heading and escape taxis altogether (Fig 6A; $D_r$ = 0.1 – 10, $H$ = 0.99). Others gradually reoriented due to stabilized persistence. As $f_{tax}$ increased, taxis overcame intrinsic persistence in a growing fraction of cells, reducing heterogeneity in final positions (Fig 6B-C; $D_r$ = 0.1 – 10, $H$ = 0.99). Fig 7B confirms that high $H$ values produced broad distributions at low $f_{tax}$, but variability diminished at high $f_{tax}$. Notably, at high $H$ and strong taxis, overshooting became common due to the combined effects of long, persistent steps and delayed reorientation, followed by eventual stabilization toward the organizing center.

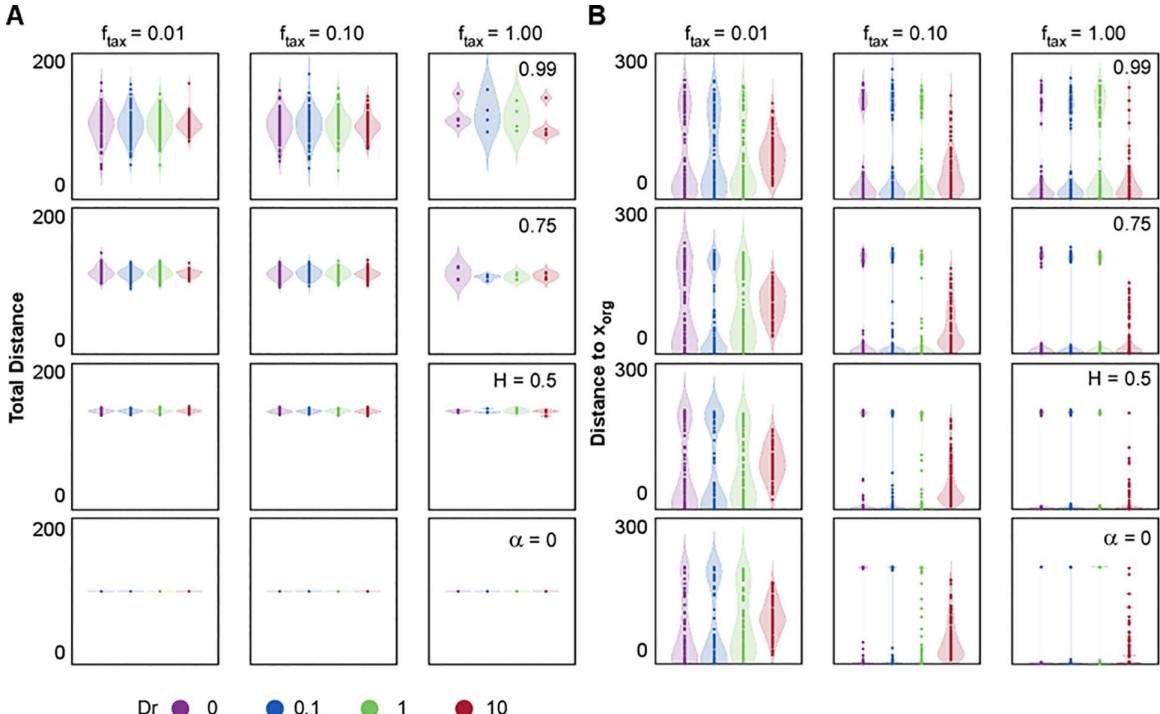

**Fig 7. Long scale quantifications of taxis model. Distributions of (A) Total distance traveled and (B) Final distance to $x_{org}$ across different values of H and Dr**. From top to bottom, each plot corresponds to a decreasing H value, while colours represent different Dr levels. In each figure from left to right the magnitude of $f_{tax}$ progressively increases. Dot plots overlaid on violin plots depict the distribution of outcomes across simulations.

Our model highlights that successful discovery of the organizing center depends on how effectively cells balance internal motion dynamics with external guidance cues. Taxis provides a directional force that orients migration toward the target, but this mechanism is modulated by intrinsic stochastic properties. High translational correlation ($H$) stabilizes initial trajectories, reducing responsiveness to weak guidance and making it more likely for cells to overshoot the target. Conversely, angular diffusion ($D_r$) plays a biphasic role in target discovery: at low levels, $D_r$ facilitates efficient reorientation, improving the probability of aligning with the organizing center. However, at high $D_r$, directionality is compromised by erratic turning, leading to scattered trajectories and poor convergence. These results show that target discovery is governed by a delicate balance between taxis and noise-driven persistence, with optimal navigation emerging from moderate flexibility in reorientation.

## Discussion

Our model reveals how two stochastic mechanisms: translational random motility with temporal autocorrelation and angular diffusion, interact to shape the stability and adaptability of migratory persistence. By introducing fBm into the translational component of cell motion, we capture temporal correlations characteristic of the stabilizing influence of persistent subcellular structures, which operate over finite timescales and define memory in the direction of motion (Figs 1-5). This temporal correlation defined by the Hurst index ($H$) imposes a mechanical constraint that preserves directionality and increment behavior over time. In contrast, angular diffusion ($D_r$) introduces reorientations that destabilize trajectories, reflecting intrinsic fluctuations such as cytoskeletal remodeling or polarity loss.

Crucially, efficient spatial exploration does not emerge from one mechanism alone, but from their balance. Under pure cell intrinsic conditions, high values of $H$ produce persistent motion, yielding long and persistent migratory tracks that are largely insensitive to the diffusion mechanism. On the other hand, at high $D_r$, trajectories become more exploratory but lose coherence, revolving continuously around the center. Only at intermediate regimes, where moderate correlation is combined with controlled angular flexibility, do we observe behaviors that resemble effective cell navigation: persistent yet responsive [59,60,70]. At this intermediate regime cells are capable of both overcoming and following external cues, as observed in experimental setups [20,32]. Interestingly, the quantifications of the MSD evolution show that anomalous superdiffusive behaviours emerge irrespective of the type and magnitude of correlation when in absence of rotational reorientation ($D_r$ = 0), aligned with reported traces of *in vitro* experimentations [64]. Our model assumes the migratory machinery as constantly turned on, with a stable lamellipodia-driven polarity when $D_r$ = 0 that still affects the displacement, causing a constant increment leading to a ballisitic process. On the other hand, when positive correlation is introduce into the system at an interplay with reorientation-driven convection, we observe a switch from diffusive to superdiffusive behaviors, regulated by the magnitude of positive correlation ($H > 0.5$). While we have not performed this analysis, we expect that subdiffusive behaviours can be obtained when in presence of negative correlation (by setting $H < 0.5$), or by switching the Brownian noise in the reorientation equation (by modulating $H_\theta$). It remains to be seen how the interplay of two correlated stochastic modules can influence the diffusive phenotype of migratory cells. For example, subdiffusive traces obtained from positively correlated modules could represent the observed *in vitro* phenotype that characterizes intermittent trapping associated with ECM physicochemical cues, such as substrate relaxation in cancer or stress relaxation in brain tissue [64].

Using our computational framework, we implemented a simple guiding cue and showed that this theme becomes especially evident under directional guidance (Figs 6-7). When external cues are introduced, translational correlation imposes inertia: cells resist reorientation and, under weak taxis, may fail to find the target and migrate seemingly unaffected by the attractant. At the same time, the fBm allows some cells to overshoot and re-engage, producing complex spatial distributions. Angular diffusion, meanwhile, plays a biphasic role: too little prevents correction and locks cells into the taxis regime, whereas too much erases directionality and introduces instability, causing cells to disperse away from the organizing center.

Together, these dynamics reflect a broader principle: successful migration requires both stability and plasticity, and their interplay defines the space of migratory strategies available to a cell. This principle extends beyond our specific framework and points to a general mechanism through which cells balance robustness and adaptability. Our findings provide a quantitative framework for understanding how cellular behaviors emerge from the interaction of stochastic noise and temporally correlated translational dynamics. This is particularly relevant for systems where persistent migration is essential, such as immune surveillance, wound healing, or cancer invasion, and where tuning the balance between exploration and commitment to specific biological programs may determine functional outcomes.

Future extensions of this work will examine the interplay of these mechanisms with (a) the mechanical forces that arise in collective cell motion and (b) migration on irregular (curved) substrates. The role of polarity and angular diffusion in collective systems has been investigated previously [31,34], but to our knowledge no study has analyzed the mechanisms considered here in experimental models of collective motility. Recent studies have also shown that substrate curvature can strongly influence cell migration, giving rise to a behavior known as *curvotaxis*. For instance, He and Jiang [71] employed a three-dimensional mechanical model to show that cells migrate more persistently along concave surfaces and display reduced motility on convex ones. This curvature-dependent migration was further validated experimentally by Pieuchot *et al.* [72], who observed that specific cell lines preferentially move toward concave valleys and avoid convex ridges, a process mediated by coupling between the actin cytoskeleton and the nucleus. Exploring how the balance between angular diffusion and persistence arising from temporal correlations, as described in this work, integrates with these curvature-driven mechanisms represents an important avenue for future research.

## Materials and methods

All codes and models were implemented using MATLAB R2023a. The equations of motion were solved directly using an implicit Euler method [73]. $\Delta t$ was chosen as 0.1 for all conditions, and 200 replicates were made for each main model and 100 for each taxis model.

## Significance Statement

Cell migration drives key biological processes such as immune surveillance, development, and cancer invasion. Most models reduce motility to random walk dynamics, overlooking temporal correlations that arise from intrinsic cellular processes. By integrating fractional Brownian motion into agent-based modeling, we show how correlated translational noise interacts with angular diffusion to produce emergent behaviors, including overshooting, exploratory loops, and persistent trajectories. Our framework unifies these outcomes under a single mechanistic description and highlights how intrinsic noise modulates taxis, exploration, and persistence. This approach provides mathematicians and cell biologists with a versatile tool to test how cells balance stability and adaptability in dynamic environments.

## Supporting information

**S1 File. Supplementary Material.** Description of Parameter Definitions and Model Solver and Timestep Limitations. (DOCX)

## Author contributions

**Conceptualization:** Ignacio Montenegro-Rojas, Diego Castelli-Lacunza, Anastasios Matzavinos, Andrea Ravasio.

**Data curation:** Ignacio Montenegro-Rojas, Martín Andaur-Lobos, Karol Soler-Orozco, Anastasios Matzavinos, Andrea Ravasio.

**Formal analysis:** Ignacio Montenegro-Rojas, Martín Andaur-Lobos, Karol Soler-Orozco, Diego Castelli-Lacunza, Anastasios Matzavinos, Andrea Ravasio.

**Funding acquisition:** Cristina Bertocchi, Anastasios Matzavinos, Andrea Ravasio.

**Methodology:** Ignacio Montenegro-Rojas, Diego Castelli-Lacunza, Anastasios Matzavinos, Andrea Ravasio.

**Software:** Ignacio Montenegro-Rojas, Martín Andaur-Lobos, Karol Soler-Orozco, Diego Castelli-Lacunza, Anastasios Matzavinos.

**Supervision:** Ignacio Montenegro-Rojas, Cristina Bertocchi, Andrea Ravasio.

**Visualization:** Anastasios Matzavinos, Andrea Ravasio.

**Writing – original draft:** Ignacio Montenegro-Rojas, Cristina Bertocchi, Anastasios Matzavinos, Andrea Ravasio.

**Writing – review & editing:** Ignacio Montenegro-Rojas, Cristina Bertocchi, Anastasios Matzavinos, Andrea Ravasio.

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
