## [Decision Letter · Decision Letter 0]

6 Jan 2026

PONE-D-25-64287MODELING CELL MIGRATORY PERSISTENCE THROUGH TEMPORAL CORRELATIONS AND ANGULAR NOISEPLOS One

Dear Dr. Ravasio,

Thank you for submitting your manuscript to PLOS ONE. After careful consideration, we feel that it has merit but does not fully meet PLOS ONE’s publication criteria as it currently stands. Therefore, we invite you to submit a revised version of the manuscript that addresses the points raised during the review process.

We look forward to receiving your revised manuscript.

Kind regards,

Haroldo V. Ribeiro

Academic Editor

PLOS One

Journal Requirements:

“This work was funded by ANID FONDECYT Regular 1210872, 1250073, 1221220; ANID FONDEQUIP EMQ210101, Núcleo Milenio SELFO NCN2024_068.

AR and AM are grateful to PUC/VRI, PUC IIBM, PUC IMC, and the graduate program at IIBM for the seed funding and support.”

4. Please note that your Data Availability Statement is currently missing the repository name. If your manuscript is accepted for publication, you will be asked to provide these details on a very short timeline. We therefore suggest that you provide this information now, though we will not hold up the peer review process if you are unable.

Additional Editor Comments:

Please submit a revised version that responds to the reviewers by improving the literature context, clarifying the model's distinct contribution and scope, and providing quantitative evidence of anomalous diffusion (including MSD and related diagnostics).

Reviewer's Responses to Questions

**Comments to the Author**

1. Is the manuscript technically sound, and do the data support the conclusions?

Reviewer #1: Yes

Reviewer #2: Yes

2. Has the statistical analysis been performed appropriately and rigorously? 

Reviewer #1: Yes

Reviewer #2: Yes

3. Have the authors made all data underlying the findings in their manuscript fully available?

Reviewer #1: Yes

Reviewer #2: Yes

4. Is the manuscript presented in an intelligible fashion and written in standard English?

Reviewer #1: Yes

Reviewer #2: Yes

5. Review Comments to the Author

Reviewer #1: The Authors use the FBM to describe cell migration by analysis of temporal correlations and angular reorientation. The presented material is interesting for the journal readers and worth considering. However, some points should be clarified.

- Appropriate citations to the FBM are missing. FBM was formulated by Kolmogorov in 1940 and around 30 years later was rediscovered by Mandelbrot and Van Ness. This should be mentioned.

- Many other papers on FBM and its applications are missing. There are many related papers on the topic by Metzler, Chechkin, Cherstvy, Sokolov, Vojta, Janušonis, Ślęzak, Krapf, etc.

- What is different in the current model. Some comparison to the other FBM models should be discussed, advantages, disadvantages, etc.

- Many sentences and statements are copied form the paper “On the role of fractional Brownian motion in models of chemotaxis and stochastic gradient ascent” by the same group of authors.

- More physical explanation and justification of the use of FBM to describe cell migration is needed. There are many other models which share same/similar properties, such as fractional Langevin equation, generalized Langevin equation, continuous time random walk, etc. Why FBM is appropriate model for the current system? Have the authors verified that the trajectories, the probability density, the velocity autocorrelation function, the MSD are adequately explained by the FBM?

- Subdiffusion and superdiffusion are mentioned in the paper, but never described. This should be explained in order to be clear for most of the readers, not just for the experts in the field.

- The “memory effects” mentioned in the paper are not adequately explained. What is the reason of the memory effects, how they are incorporated in the model, etc?

- Can you show the anomalous diffusive character of the process of interest by presenting results for the MSD (ensemble averaged/time averaged MSD)?

Reviewer #2: The authors propose a model to describe cell mobility. This new model generalized the previous one by considering fractional Brownian motion with persistent, correlated noise instead of Gaussian noise. The model consists of two main stochastic equations: a Langevin-like equation to describe the motion (where there is friction and a correlated noise), and the equation that describes how the angle varies according to time, proportional to the square of the diffusion coefficient and the correlated noise (which replaces the Gaussian noise). The authors assume some 'postulates' such as the non-concern of the energetic states (where the energy comes from and how it is expended) and that, for biological reasons, the fractional Brownian motion is only persistent.

The simulations and results are well-presented, clearly illustrating the interplay between these two mechanisms of anomalous diffusion. After the revision the suggestion is to accept the manuscript by considering minor revisions that would improve it:

Revisions suggested:

1) In the introduction, models of the literature with the interplay of mechanisms of anomalous diffusion could be discussed by the authors. The complexity of living systems cannot (to date) be described by only one mechanism of anomalous diffusion. Over the last 20 years, several models have demonstrated that the interplay of mechanisms yields non-trivial behaviors.

2) It is not explicity if the models cause supperdifusion or subdiffusion. The authors mention the MSD in the introduction, but the analysis does not follow such an approach to description. It should be interesting to find that such an interplay of persistent, correlated noise can lead to subdiffusion, which is not a trivial case. Therefore, the MSD calculation could "talk" easier with the community of the anomalous diffusion research and enhance the findings presented here.

3) In the case of anomalous behaviors such subdiffusion, the author should research about the advantage of cells or living diffusion be subdiffusive in the case of targeting search.

6. PLOS authors have the option to publish the peer review history of their article (what does this mean?). If published, this will include your full peer review and any attached files.

Reviewer #1: No

Reviewer #2: **Yes:** Angel Akio Tateishi

---

## [Author Response · Author response to Decision Letter 1]

13 Apr 2026

A specific file has been uploaded. Here is the content of that response letter.

Reviewer #1: The Authors use the FBM to describe cell migration by analysis of temporal correlations and angular reorientation. The presented material is interesting for the journal readers and worth considering. However, some points should be clarified.

We thank the reviewer for their constructive and thoughtful evaluation of our work. We appreciate their recognition of the relevance of the study and their insightful comments, particularly regarding the quantitative characterization of anomalous diffusion and the clarification of key concepts. Addressing these points has strengthened the manuscript by improving the rigor of the analysis and the clarity of the conceptual framework. For clarity, we have grouped related comments and structured our responses accordingly, addressing each point in detail below.

Fractional Brownian Motion: citations, context, and positioning

- Appropriate citations to the FBM are missing. FBM was formulated by Kolmogorov in 1940 and around 30 years later was rediscovered by Mandelbrot and Van Ness. This should be mentioned.

- Many other papers on FBM and its applications are missing. There are many related papers on the topic by Metzler, Chechkin, Cherstvy, Sokolov, Vojta, Janušonis, Ślęzak, Krapf, etc.

We thank the reviewer for this valuable comment. We have revised the Introduction to mention the original formulation of fractional Brownian motion by Kolmogorov (1940), as well as its subsequent development by Mandelbrot and Van Ness (1968), thereby providing appropriate historical context. In addition, we have incorporated a broader set of references covering key contributions to the theory and applications of anomalous diffusion and fractional Brownian motion, including works by Metzler, Mishura, Chechkin, Cherstvy, Nourdin, Sokolov, Vojta, Janušonis, Ślęzak, Krapf, and others (Lines 130–133; Line 136).

These additions have also allowed us to broaden the contextual framing of our work within the landscape of anomalous diffusion models. We have accordingly expanded the Discussion to better position our approach and its scope (Lines 472–481).

- What is different in the current model. Some comparison to the other FBM models should be discussed, advantages, disadvantages, etc.

We appreciate this request from the reviewer, as it allowed us to expand on our rationale behind the use of fBm as a model for correlation of motion. Regarding the choice of fBm over alternative formulations, we noted that these approaches typically incorporate memory through explicit integral kernels, viscoelastic response functions, or waiting-time distributions, which introduce additional parameters and physical assumptions that are not trivial to relate biologically in the context of cell migration. In our work, fBm introduces long-range temporal correlations through a single parameter (H), while preserving Gaussian statistics and stationary increments. We have expanded on this rationale in the Introduction (Lines 133-140), where we highlight the non-Markovian nature of the correlated increments.

- Many sentences and statements are copied form the paper “On the role of fractional Brownian motion in models of chemotaxis and stochastic gradient ascent” by the same group of authors.

We apologize if the manuscripts appeared similar in parts. We would like to respectfully clarify that any overlap did not arise from intentional reuse of text. As both manuscripts are developed within a related theoretical setting and by the same group of authors, some similarity in the presentation of background concepts may occur. However, we agree that the phrasing should be clearly differentiated, and we have carefully revised the manuscript to eliminate textual overlap and reduce repetition, particularly in the Introduction and in the description of fractional Brownian motion, while preserving clarity for readers unfamiliar with the related work.

- More physical explanation and justification of the use of FBM to describe cell migration is needed. There are many other models which share same/similar properties, such as fractional Langevin equation, generalized Langevin equation, continuous time random walk, etc. Why FBM is appropriate model for the current system?

In line with the comment above regarding the rationale behind fBm, we have expanded our discussion (lines 472-481) and explanation of fBm and its capabilities, focusing on why it is adequate to be applied in cellular systems, specifically due to its ability to introduce long-range temporal correlations through a single parameter (H), making it a minimal yet mechanistically grounded framework.

Verification of anomalous diffusion via MSD analysis and clarification of sub- and superdiffusive behavior

We thank the reviewer for these important and insightful comments. This analysis has significantly strengthened the manuscript by enabling a more quantitative characterization of the model and by allowing clearer conclusions to be drawn regarding the diffusive behavior of the system. We address the specific points below.

(i) Verification of trajectories and diffusive behavior.

- Have the authors verified that the trajectories, the probability density, the velocity autocorrelation function, the MSD are adequately explained by the FBM?

To assess whether the model captures the expected properties of correlated stochastic motion, we have incorporated a quantitative analysis of the Mean Square Displacement (MSD) in the revised manuscript (Fig. 5; Lines 345–377). The MSD is computed as the ensemble average of the squared displacement from the origin across simulated trajectories. In the absence of angular reorientation (Dr=0), we observe ballistic behavior across all tested conditions. The magnitude of translational correlation (controlled by H) can switch the behavior from diffusive to superdiffusive, with notable perturbations due to the adjection term regulated by the polarity angle and motile force Fm. As angular diffusion increases (Dr>0), differences between conditions become more apparent: highly correlated trajectories (H≈0.99) maintain superdiffusive behavior, whereas lower correlation reduces spreading, though still deviating from purely Brownian dynamics. These results demonstrate that the trajectories generated by the model exhibit consistent anomalous diffusion properties arising from the interplay between translational correlation and angular noise.

(ii) Clarification of subdiffusion and superdiffusion.

- Subdiffusion and superdiffusion are mentioned in the paper, but never described. This should be explained in order to be clear for most of the readers, not just for the experts in the field.

We have clarified the meaning of subdiffusive and superdiffusive behavior in the Introduction (Lines 98–116). Specifically, subdiffusion is defined as motion with hindered spreading over time, whereas superdiffusion corresponds to enhanced spreading, typically associated with persistent, directed motion. These definitions are now directly connected to the MSD analysis, where the diffusive regimes are quantitatively characterized through the temporal scaling of the displacement.

(iii) Demonstration of anomalous diffusion via MSD.

- Can you show the anomalous diffusive character of the process of interest by presenting results for the MSD (ensemble averaged/time averaged MSD)?

In response to the reviewer’s request, we have added a dedicated MSD analysis (Fig. 5; Lines 345–377), which explicitly demonstrates the anomalous diffusive character of the system. We further evaluated the stability of the MSD by varying the number of sampled trajectories, confirming that the observed behavior is robust when considering sufficiently large ensembles. This addition strengthens the quantitative support for the role of temporal correlations in shaping migratory dynamics.

Finally, we have expanded the Discussion to contextualize these findings (Lines 472–481), highlighting that superdiffusive behavior emerges robustly across the explored parameter space and outlining how alternative regimes (e.g., subdiffusion) may arise under different correlation structures.

Clarification of “memory effects”: biological origin and model implementation

- The “memory effects” mentioned in the paper are not adequately explained. What is the reason of the memory effects, how they are incorporated in the model, etc?

We thank the reviewer for raising this point. We have revised the manuscript to clarify the meaning of memory and its role in the model (lines 107–116). While this terminology is commonly used in the fractional Brownian motion literature to denote temporal correlations, we recognize that in a multidisciplinary context it may be interpreted in different ways, particularly by a biomedical audience. To address this, we now define memory operationally in the Introduction as temporal correlations in cell displacement, arising from the finite timescales associated with the assembly and disassembly of subcellular structures that sustain directed migration (e.g., lamellipodia, adhesion complexes, and actin networks). These processes stabilize cell polarity over short time windows and introduce persistence in motion. In the model, this effect is incorporated through fractional Brownian motion, where the Hurst exponent controls the degree of temporal correlation in the stochastic increments. In addition, we have tempered the use of the term memory throughout the manuscript to improve precision and avoid ambiguity.

Reviewer #2: The authors propose a model to describe cell mobility. This new model generalized the previous one by considering fractional Brownian motion with persistent, correlated noise instead of Gaussian noise. The model consists of two main stochastic equations: a Langevin-like equation to describe the motion (where there is friction and a correlated noise), and the equation that describes how the angle varies according to time, proportional to the square of the diffusion coefficient and the correlated noise (which replaces the Gaussian noise). The authors assume some 'postulates' such as the non-concern of the energetic states (where the energy comes from and how it is expended) and that, for biological reasons, the fractional Brownian motion is only persistent.

The simulations and results are well-presented, clearly illustrating the interplay between these two mechanisms of anomalous diffusion. After the revision the suggestion is to accept the manuscript by considering minor revisions that would improve it.

Revisions suggested:

1) In the introduction, models of the literature with the interplay of mechanisms of anomalous diffusion could be discussed by the authors. The complexity of living systems cannot (to date) be described by only one mechanism of anomalous diffusion. Over the last 20 years, several models have demonstrated that the interplay of mechanisms yields non-trivial behaviors.

We thank the reviewer for this important point. We agree that the complexity of living systems is not captured by a single mechanism of anomalous diffusion, and that the interplay between multiple processes can give rise to non-trivial behaviors.

Over the past thirty years, several computational models have examined stochastic mechanisms underlying motility in prokaryotic cells and in selected eukaryotic systems with well-defined movement patterns, such as slime mold amoebae. At the single-cell level, stochastic models for bacterial run-and-tumble dynamics have been proposed, including velocity-jump processes and, more generally, processes with velocity redistribution kernels (Stevens, 1997; 2000; Erban & Othmer, 2001). Likewise, reinforced random walks and related models have often been used for slime mold amoebae such as Dictyostelium. However, to the best of our knowledge, most of the computational literature on eukaryotic cell motility, with the notable exception of Dictyostelium, still relies heavily on Brownian motion to model random exploration. In response, we have expanded the Introduction to better reflect this perspective, incorporating a broader discussion of existing models and their respective roles in describing anomalous transport in biological systems. These additions provide context for our approach and clarify how our framework relates to and complements previous studies (Lines 117–116; 130–141).

2) It is not explicity if the models cause supperdifusion or subdiffusion. The authors mention the MSD in the introduction, but the analysis does not follow such an approach to description. It should be interesting to find that such an interplay of persistent, correlated noise can lead to subdiffusion, which is not a trivial case. Therefore, the MSD calculation could "talk" easier with the community of the anomalous diffusion research and enhance the findings presented here.

We thank the reviewer for this insightful suggestion. In response, we have incorporated a detailed analysis of the Mean Square Displacement (MSD) in the revised manuscript (Fig. 5; Lines 345–377), allowing us to explicitly characterize the diffusive behavior emerging from the model. The MSD is computed as the ensemble average of the squared displacement from the origin across simulated trajectories, enabling us to assess the temporal scaling of motion and distinguish between subdiffusive and superdiffusive regimes.

Our analysis shows that, in the absence of angular reorientation (D_r=0), the system exhibits superdiffusive behavior across all tested conditions, with MSD scaling nonlinearly in time. The magnitude of translational correlation (controlled by H) primarily affects the scale and persistence of displacement rather than the qualitative diffusive regime. As angular diffusion increases (D_r>0), differences between conditions become more pronounced: highly correlated trajectories (H≈0.99) maintain superdiffusive behavior, whereas lower correlation leads to reduced spreading, though still deviating from purely Brownian dynamics. These results highlight how the interplay between correlated translational dynamics and angular noise governs the effective exploration of space.

In addition, we have expanded the Discussion to address the non-trivial behaviors arising from this interplay (Lines 472–481). In particular, we outline how alternative regimes, such as subdiffusion, may emerge under different correlation structures, for example by introducing correlated noise in the reorientation dynamics. These additions strengthen the connection of our work to the anomalous diffusion literature and enhance the interpretation of the results.

3) In the case of anomalous behaviors such subdiffusion, the author should research about the advantage of cells or living diffusion be subdiffusive in the case of targeting search.

It is important to note that, within the parameter space explored in this study, we do not observe subdiffusive regimes, as the system consistently exhibits superdiffusive behavior. However, we agree that subdiffusive dynamics can play an important role in biological search processes. Accordingly, we have expanded the Introduction to include a discussion of the functional implications of anomalous diffusion in biological systems, including how subdiffusive behaviors may arise and contribute to search efficiency in structured or heterogeneous environments (Lines 98-104). In particular, we highlight how hindered spreading and intermittent exploration can be advantageous under conditions of confinement or trapping. We have complemented this discussion with additional references and examples linking anomalous diffusion to cellular migration (Lines 104–112), thereby providing broader biological context for our study. Within this framework, our results provide a quantitative description of how temporal correlations influence migratory dynamics, while also outlining how alternative regimes may emerge under different correlation structures, for example through negative temporal correlations or correlated reorientation dynamics. These additions clarify the biological relevance of anomalous diffusion in our model and outline potential directions for future work, including experimental validation and extensions to multicellular systems with cell–cell interactions

---

## [Decision Letter · Decision Letter 1]

30 Apr 2026

MODELING CELL MIGRATORY PERSISTENCE THROUGH TEMPORAL CORRELATIONS AND ANGULAR NOISE

PONE-D-25-64287R1

Dear Dr. Ravasio,

We’re pleased to inform you that your manuscript has been judged scientifically suitable for publication and will be formally accepted for publication once it meets all outstanding technical requirements.

Kind regards,

Haroldo V. Ribeiro

Academic Editor

PLOS One

Additional Editor Comments (optional):

Both reviewers confirm that you have addressed their previous concerns and that the additions, particularly the mean-squared-displacement analysis, have substantially strengthened the manuscript. I am therefore pleased to recommend it for publication in PLOS ONE and congratulate you on this excellent work.

Reviewers' comments:

Reviewer's Responses to Questions

**Comments to the Author**

1. If the authors have adequately addressed your comments raised in a previous round of review and you feel that this manuscript is now acceptable for publication, you may indicate that here to bypass the “Comments to the Author” section, enter your conflict of interest statement in the “Confidential to Editor” section, and submit your "Accept" recommendation.

Reviewer #1: All comments have been addressed

Reviewer #2: All comments have been addressed

2. Is the manuscript technically sound, and do the data support the conclusions?

Reviewer #1: Yes

Reviewer #2: Yes

3. Has the statistical analysis been performed appropriately and rigorously? 

Reviewer #1: Yes

Reviewer #2: Yes

4. Have the authors made all data underlying the findings in their manuscript fully available?

Reviewer #1: Yes

Reviewer #2: Yes

5. Is the manuscript presented in an intelligible fashion and written in standard English?

Reviewer #1: Yes

Reviewer #2: Yes

6. Review Comments to the Author

Reviewer #1: The manuscript has been improved and the manuscript can be accepted for publication. The added part with analysis of the mean squared displacement significantly support the findings in the manuscript.

Reviewer #2: The authors have addressed the concerns and suggestions raised in my first evaluation. My review of the revised version was thus pragmatic in scope, focusing primarily on the issues previously identified and on whether they had been satisfactorily resolved.

7. PLOS authors have the option to publish the peer review history of their article (what does this mean?). If published, this will include your full peer review and any attached files.

Reviewer #1: No

Reviewer #2: No

---

## [Editor Report · Acceptance letter]

PONE-D-25-64287R1

PLOS One

Dear Dr. Ravasio,

I'm pleased to inform you that your manuscript has been deemed suitable for publication in PLOS One. Congratulations! Your manuscript is now being handed over to our production team.

Kind regards,

on behalf of

Dr. Haroldo V. Ribeiro

Academic Editor

PLOS One